# Accurate Detection of *Salmonella* Based on Microfluidic Chip to Avoid Aerosol Contamination

**DOI:** 10.3390/foods11233887

**Published:** 2022-12-01

**Authors:** Yining Luo, Shan Shan, Shuanglong Wang, Jinlin Li, Daofeng Liu, Weihua Lai

**Affiliations:** 1State Key Laboratory of Food Science and Technology, Nanchang University, 235 East Nanjing Road, Nanchang 330047, China; 2College of Life Science, National R&D Center for Freshwater Fish Processing, Jiangxi Normal University, Nanchang 330022, China; 3Jiangxi Province Key Laboratory of Diagnosing and Tracing of Foodborne Disease, Jiangxi Province Centre for Disease Control and Prevention, 555 East Beijing Road, Nanchang 330029, China; 4Jiangxi Key Laboratory for Mass Spectrometry and Instrumentation, East China University of Technology, Nanchang 330013, China

**Keywords:** *Salmonella*, LAMP, CRISPR/Cas12a, visual detection

## Abstract

*Salmonella* is a type of common foodborne pathogen of global concern, seriously endangering human health. In molecular biological detection of *Salmonella*, the method of amplifying DNA often faces the problem of aerosol pollution. In this study, a microfluidic chip was developed to integrate loop-mediated isothermal amplification (LAMP) and clustered regularly interspaced short palindromic repeats (CRISPR)/Cas12a system to detect *Salmonella*. The LAMP reaction solution was initially injected into the chamber to amplify at 65 °C for 20 min; the CRISPR/Cas12a reaction solution was subsequently injected to mix with the amplicons for fluorescent signal production at 43 °C for 30 min. Then, the results can be confirmed by naked eyes under 495 nm light or by a fluorescence immunochromatographic reader. The detection limit of this method for *Salmonella* DNA was 118 pg/μL. The sensitivity and specificity of this method was 100%. Furthermore, this method was used to detect *Salmonella* after enrichment for 4 h in salmon and chicken samples spiked with 30 CFU/25 g, and was verified to have a stable detection capability in real samples. The microfluidic chip integrated with the LAMP and CRISPR/Cas12a system not only provides a possibility of highly sensitive endpoint fluorescent visual detection of a foodborne pathogen, but also greatly eliminates the risk of aerosol contamination.

## 1. Introduction

*Salmonella* is an important pathogen that poses a substantial threat to human health. Approximately 86% of the described salmonellosis cases were caused by foodborne infections [1]. Aquatic products, especially salmon eaten without heat treatment, are easily contaminated with *Salmonella* and more likely cause harm to human health [2]. Human infection by *Salmonella* strains may result in fever, vomiting, abdominal pain, nausea, hemorrhagic enteritis, and other symptoms [3,4,5]. Therefore, a rapid and accurate method to detect *Salmonella* contamination in food is required. 

Loop-mediated isothermal amplification (LAMP) has been developed for its simplicity, high amplification capacity, and rapidity [6,7,8]. LAMP requires Bst DNA polymerase and a set 4–6 primers, which can be completed at a constant temperature (60–65 °C) without an expensive thermocycler [9]. Furthermore, LAMP can amplify 10^9^-fold target sequence copies within 15–60 min [10,11]. The amplification capacity of LAMP is 10–100 times higher than that of traditional polymerase chain reaction (PCR) [12]. On the basis of these advantages, LAMP has been widely applied in the field of foodborne pathogen detection [13]. 

The clustered regularly interspaced short palindromic repeats (CRISPR)/Cas system can specifically bind and cleave target nucleic acids under the guidance of CRISPR RNA (crRNA), which identifies and captures target DNA by base-complementary pairing [14,15,16]. The CRISPR/Cas system has shown the potential to improve the specificity and accuracy of genetic detection [17,18]. CRISPR/Cas12a, from the type V-A CRISPR system, exhibits site-specific dsDNA cutting and nonspecific ssDNA trans-cleavage ability, which provides an efficient signal amplification tool for molecular diagnostics [19]. 

LAMP-binding CRISPR/Cas system has been applied in the field of disease diagnosis and foodborne pathogen detection [20,21,22]. The reaction process of the LAMP-binding CRISPR/Cas12a system requires two separate steps because the high temperature in the course of LAMP may reduce the activity of Cas12a. However, opening tubes and transferring LAMP solution into CRISPR/Cas reaction solution would lead to aerosol leakage [21,23]. Reports have shown that LAMP amplicons can contaminate reagents in the laboratory by aerosol for several weeks, and they can be promptly used as templates in subsequent reactions, thereby jeopardizing the results of the reactions. When contamination begins, the false-positive rate of the test results would increase sharply, unless all reagents and primers were replaced [24]. Herein, we designed a microfluidic chip as the platform to integrate LAMP and the CRISPR/Cas12a system for the accurate detection of *Salmonella*. 

## 2. Materials and Methods

### 2.1. Bacterial Strains and DNA Extraction

The standard bacterial strains and clinically isolated strains were from the Jiangxi Province Center for Disease Control and Prevention. Detailed information of all strains is shown in Table 1. All bacterial strains were cultured in trypticase soy broth (TSB, Land Bridge Technology Co., Ltd., Beijing, China) on a shaking incubator at 180 rpm and 37 °C overnight. One milliliter of each bacterial culture in TSB was centrifuged at 8000 rpm/min for 5 min. Then, the pellet was washed with sterilized 1× PBS (Solarbio Co., Ltd., Beijing, China) for three times and resuspended with 1 mL 1× PBS. Resuspension was boiled for lysing bacteria cells, and then centrifuged at 8500 rpm for 5 min. The supernatants were collected to obtain genomic DNA. The concentration of DNA was determined by Micro-Spectrophotometer K5600 (Kaiao Co., Ltd., Beijing, China). 

### 2.2. LAMP Primers, crRNA, and Report DNA Design

*invA*, a virulence gene, is chromosomally located and conserved in almost all the *Salmonella* serotypes [25,26]. The sequence of *invA* (GenBank No. M90846.1) was used to design LAMP primers. LAMP primers, including the forward inner primer (FIP), the backward inner primer (BIP), the outer forward primer (F3), the outer backward primer (B3), and additional loop primers (FL and BL) were designed via Primer Explorer software Version 5 (http://primerexplorer.jp/lampv5e/index.html accessed on 20 June 2021). The crRNA for specifically recognizing the specific amplicon was designed through the Benchling website (https://www.benchling.com/crispr/ accessed on 3 July 2021). Report DNA was designed as a short ssDNA (5 nt), labeled with fluorophore (6-FAM) and quencher (BHQ1) groups at both ends [27]. The primers, crRNA, and report DNA were synthesized by Sangon Co., Ltd. (Shanghai, China); the detailed sequence information is listed in Table 2.

### 2.3. LAMP Assay

LAMP solution was prepared according to New England Biolabs (NEB) instruction in a total volume of 25 μL, containing 3.5 μL of 10 mM dNTP mix (Vazyme Co., Ltd., Nanjing, China), 2.5 μL of 10× isothermal amplification buffer, 1.5 μL of 100 mM MgSO_4_, 1 μL of 800 U/mL Bst 2.0 DNA polymerase (New England Biolabs, Ipswich, MA, USA), 1 μL of 40 μM FIP/BIP primers, 1 μL of 5 μM F3/B3 primers, 1 μL of 10 μM FL/BL primers, and 2 μL DNA template. Then, 13 μL of mixture was incubated in thermostatic metal bath TU-10 (Yiheng Co., Ltd., Shanghai, China) at 65 °C for 5, 10, 15, 20, and 25 min and terminated at 80 °C for 10 min. The product of LAMP was characterized by QIAxcel advanced capillary electrophoresis (Qiagen, Hilden, Germany). 

### 2.4. LAMP–CRISPR/Cas12a Detection System

The LAMP assay was performed according to Section 2.3. The Cas12a-mediated detection system contained 1.95 μL 10× NEBuffer 2.1, 1.3 μL of Cas12a (1, 2, 3, and 4 μM) (New England Biolabs, Ipswich, MA, USA), 1.95 μL of crRNA (0.5, 1.0, 1.5, 2.0, and 2.5 μM), 1.3 μL of report DNA (20, 40, 60, 80, and 100 μM), and 13 μL of LAMP amplicons as an activator. Then, 19.5 μL of mixture was incubated a thermostatic metal bath at 25 °C, 31 °C, 37 °C, 43 °C, and 49 °C for 30 min. The products of LAMP cleaved by Cas12a were characterized by capillary electrophoresis.

### 2.5. Detection of Salmonella by Microfluidic Chip Integrated with LAMP and CRISPR/Cas12a System

The polydimethylsiloxane (PDMS) chip was designed to be a reaction and observation platform, with one observation chamber and two narrow channels for injection of reaction solution. Thirteen microliters of LAMP components were initially injected into the chamber by microinjectors (Anting Co., Ltd., Shanghai, China) with smart syringe pump XMSP-C (Ximai Co., Ltd., Nanjing, China) at the speed of 15 μL/min. Then, the chip was stored at 65 °C for amplification, and subsequently at 80 °C for enzyme inactivation. After amplification, 6.5 μL of CRISPR/Cas12a components were injected into the chamber at the speed of 15 μL/min to mix with LAMP amplicons by another channel, and then the chip was stored at 43 °C. The fluorescence could be observed by the naked eyes under 495 nm light and measured by a fluorescence immunochromatographic reader (Helmen Co., Ltd., Hangzhou, China). Actual devices and the effect diagrams are shown in Appendix A. 

### 2.6. Evaluation of Detection Limit

Typically, the detection limit is determined by setting a threshold, which can be calculated from the average fluorescence intensity and standard deviation of the negative control. The value of the threshold equals the average intensity and three times the standard deviation [28].

Extracted genomic DNA that served as the template was performed a 10-fold gradient dilution from 10^−2^ to 10^−4^ with 1× PBS to evaluate the detection limit of this method. The DNA templates of each concentration were amplified by LAMP. Then, amplicons were employed to activate the CRISPR/Cas12a cleaving system. The fluorescence intensity of these results was measured by a fluorescence immunochromatographic reader.

### 2.7. Sensitivity and Specificity

One *Salmonella* standard strain, twelve *Salmonella* clinical strains, nine non-*Salmonella* standard strains, and three non-*Salmonella* clinical strains were selected to test the sensitivity and specificity of this method. DNA extracted from these strains was amplified by LAMP. Then, amplicons were employed for activating the CRISPR/Cas12a cleaving system. The fluorescence intensity of these results was measured by a fluorescence immunochromatographic reader.

### 2.8. Detection of Salmonella in Artificially Contaminated Salmon and Chicken

Salmon and chicken samples were purchased from a local supermarket. Twenty-five grams of salmon and chicken samples, which were verified to be *Salmonella*-free, were made into homogenates and spiked with *Salmonella* at a final concentration of 30 CFU/25 g, whereas un-spiked samples were tested to evaluate the specificity of this method in real samples [29]. Then, samples were placed into TSB media (225 mL of each) and incubated at 37 °C with shaking at 180 rpm. Two milliliters of culture solution were aspirated every hour from 0 to 6 h. Solutions collected were divided into two groups. One group was tested by the LAMP–CRISPR/Cas12a system, and another group was used for colony count by *Salmonella* chromogenic medium (CHRO Magar, Paris, France).

To check the effect of the matrix of real samples on the test efficiency in comparing with the buffer solution, 25 g of salmon and chicken samples were made into homogenates with TSB medium, and all samples were irradiated under a UV lamp (15 W) for 30 min to ensure sterility. Preprocessed samples and the other TSB medium with no sample were spiked with *Salmonella* at final concentrations of 4 × 10^1^, 4 × 10^2^, 4 × 10^3^, 4 × 10^4^ CFU/mL, respectively, and then tested by the LAMP–CRISPR/Cas12a system.

## 3. Results

### 3.1. Principle and Operation of the Detection System

The principle and operation of *Salmonella* detection by the microfluidic chip integrated with the LAMP and CRISPR/Cas12a system are shown in Figure 1. In summary, crude genomic DNA of *Salmonella* extracted using the heated lysis method was rapidly amplified by LAMP. The amplicons could be recognized and captured by crRNA, and then the CRISPR/Cas12a system could cleave LAMP amplicons with site-specific dsDNA cutting ability and report DNA with nonspecific ssDNA trans-cleavage ability. The cleavage of report DNA, which was labeled with fluorophore 6-FAM and quencher BHQ1, resulted in the appearance of fluorescence, as shown in Figure 1A.

PDMS chip was selected as the detection platform to eliminate the risk of aerosol contamination. LAMP components with or without DNA of *Salmonella* were initially injected into the chamber through one channel, and CRISPR/Cas12a components were injected into the chamber through another channel after LAMP was completed, as shown in Figure 1B. Then, the result could be confirmed by the naked eyes under 495 nm light or by a fluorescence immunochromatographic reader because of the suitable size of chip pre-designed. 

The risk of aerosol leakage was eliminated in this process owing to the leakproofness of the chip, thereby protecting the detection environment and reagents from aerosol pollution. With this chip, the false-positive results are greatly avoided, providing a platform for more accurate detection results in subsequent detection. 

### 3.2. Analysis of LAMP–CRISPR/Cas12a Detection System

The extracted genomic DNA of *Salmonella* was amplified by LAMP. The capillary electrophoresis image (Appendix A) showed the bands, which were the LAMP product and the Cas12a-cleaved LAMP product. The difference between the two sets of bands indicated the occurrence of the cleavage. 

LAMP amplicons were employed for activating the CRISPR/Cas12a cleaving system. More LAMP amplicons were contributed to obtain a higher fluorescence intensity of the CRISPR/Cas12a detection system. After 20 min, the LAMP amplification system generated ample target molecules for the CRISPR/Cas12a detection system (Figure 1). The optimal reaction time of LAMP to improve the detection efficiency was 20 min.

The experimental factors, including the concentration of CRISPR/Cas12a components and reaction temperature, were investigated to reduce the detection limit. First, the fluorescence intensity increased as the concentration of Cas12a increased, as shown in Figure 2A. When the concentration of Cas12a ranged 3–4 μM, the fluorescence intensity did not increase considerably. Therefore, the optimal concentration of Cas12a was 3 μM. Second, the concentration of crRNA was optimized. As the concentration of crRNA increased, the fluorescence intensity increased significantly initially, and then the increase magnitude gradually decreased, as shown in Figure 2B. Thus, 2 μM was finally selected as the optimal concentration of crRNA. Third, as the concentration of report DNA increased, the fluorescence intensity peaked at 80 μM and then decreased, as shown in Figure 2C. Hence, 80 μM was selected as the optimal concentration. Fourth, five temperatures from 25 °C to 49 °C were selected for the experiment. The fluorescence intensity peaked at 43 °C, and the fluorescence intensity began to decrease as the temperature continued to increase, as shown in Figure 2D. This result might be due to the effect of higher temperature on the activity of the enzyme. Therefore, 43 °C was selected as the optimal reaction temperature.

### 3.3. Detection Limit of the Proposed Method

After optimizing the conditions, this method was established for *Salmonella* detection. A 10-fold dilution series of extracted genomic DNA from 10^−2^ to 10^−4^ was used to determine the detection limit of this method, and the original concentration of the DNA was 118 ng/μL. The results are shown in Figure 3. The values of fluorescence intensity were 6, 162, and 232 when the concentrations of DNA were 11.8, 118, and 1180 pg/μL, respectively. The threshold was calculated as 6; thus, 118 pg/μL was regarded as the detection limit of the proposed method.

### 3.4. Sensitivity and Specificity

As shown in Figure 4, compared with the threshold (5), all twelve non-*Salmonella* strains had no evident fluorescence signal (5, 5, 5, 4, 5, 5, 4, 4, 5, 4, 5, and 5), thereby confirming that the detection method established in this study had high specificity. Meanwhile, all thirteen *Salmonella* strains had evident fluorescence signals (220, 310, 188, 200, 250, 293, 281, 277, 299, 210, 213, 230, and 259, respectively), confirming that the detection method had high sensitivity.

### 3.5. Detection of Salmonella in Salmon and Chicken Using the Proposed Method

Furthermore, the performance of the LAMP–CRISPR/Cas12a detection system was evaluated by salmon and chicken samples spiked with *Salmonella* (30 CFU/25 g), and un-spiked samples were tested to evaluate the specificity of this method in real samples. DNA extracted from bacteria with different culture times (0, 1, 2, 3, 4, 5, and 6 h) was detected by this method. As Figure 5 shows, *Salmonella* in a spiked salmon sample cultured at 4 h was detected with a fluorescence intensity of 188 when the threshold was 3, and *Salmonella* in a spiked chicken sample cultured at 4 h was detected with a fluorescence intensity of 172. The results indicated that *Salmonella* in both salmon and chicken samples could be detected by this system after only 4 h of enrichment. Appendix A shows that after 4 h of enrichment, the concentration of *Salmonella* was approximately 5 × 10^2^ CFU/mL in the salmon sample and approximately 4.2 × 10^2^ CFU/mL in the chicken sample (100 μL of culture solution was used for colony count). Meanwhile, the groups of un-spiked samples after 6 h of culture did not show positive results even with large amounts of *Escherichia coli*, indicating that this method had high specificity in real samples. 

To check the effect of the matrix of real samples on the test efficiency, TSB medium, salmon sample, and chicken sample were spiked with *Salmonella* (4 × 10^1^, 4 × 10^2^, 4 × 10^3^, and 4 × 10^4^ CFU/mL). As Figure 6 shows, 400 CFU/mL of *Salmonella* could be detected in TSB medium, salmon sample, and chicken sample with fluorescence intensities of 281, 227, and 246, respectively. The results indicated that the matrix of real sample did not affect the detection efficiency of this method, but only slightly weakened the fluorescence intensity. Therefore, this method had a stable detection capability in real samples.

### 3.6. Discussion

Due to inadequate storage temperatures or inadequate cooking, foods can be a source of pathogen infections, causing tremendous harm to human health. Therefore, pathogens including *Salmonella* should be controlled in foods. Mukama et al. [20] reported a simple, inexpensive, and ultrasensitive DNA probe based LFB with CRISPR/Cas and LAMP, which achieved high sensitivity and specificity both in pure and complex samples. However, opening tubes and transferring LAMP products into the CRISPR/Cas reaction solution would lead to aerosol leakage, which would seriously jeopardize the results of the subsequent detection. Therefore, we need a leakproof and observable reaction platform. In an attempt to establish a more rapid and accurate method to detect *Salmonella*, we developed a LAMP combined with CRISPR/Cas12a integrated into a microfluidic chip system. 

First, we optimized the reaction conditions of the LAMP and CRISPR/Cas12a system to reduce the detection limit. Then, we evaluated the performance of this method. The results showed this method had high sensitivity and specificity, which were consistent with previous reports [21]. Its excellent performance was due to two sets of pre-designed specific sequences used in the system. Six primers of LAMP made the amplification process have high sensitivity and efficiency. The designed crRNA could specifically recognize the LAMP amplicons for secondary confirmation, which further improved the accuracy of the detection. Furthermore, the performance of the LAMP–CRISPR/Cas12a detection system was evaluated by salmon and chicken samples spiked with *Salmonella*. The results showed the good detection performance for *Salmonella* in real samples, affirming the practical application potential of this method.

Further research aimed at optimizing the design and reducing the cost of the microfluidic chip may enhance the applicability of the assay. This method can also be applied to the detection of other pathogens, providing great potential as a universal platform for pathogen detection. Therefore, exploring the possibility of high-throughput detection on a single chip will also be of great interest. 

## 4. Conclusions

In summary, this study developed a LAMP combined CRISPR/Cas12a integrated into a microfluidic chip system for *Salmonella* detection. Two separate liquid additions were performed through two narrow channels to achieve two-step reactions on one chip, eliminating the risk of aerosol contamination and cross-contamination that could result from opening the cap of a centrifuge tube, which is a typical problem associated with LAMP. The designed chip can be used as a reaction platform as well as a platform for reading the results directly. The size of the chip was designed to be suitable to allow the experimental results to be read by a portable fluorescence immunochromatographic reader or directly interpreted by the naked eyes under the irradiation of a certain wavelength of ultraviolet light. The detection limit of the proposed method could reach 118 pg/μL of crude genomic DNA, and the entire detection process could be completed within 50 minutes. Furthermore, this method was used to detect *Salmonella* after enrichment for 4 h in salmon and chicken samples spiked with 30 CFU/25 g, and was verified to have a stable detection capability in real samples. At the same time, the detection method had high sensitivity and specificity for detecting twelve *Salmonella* strains and thirteen non-*Salmonella* strains. The results showed that the proposed detection system was suitable for on-site rapid *Salmonella* detection with a low detection limit. Its high performance provides great potential as a universal platform for pathogen detection.

## Data Availability

Data are contained within the article and Appendix A.

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
