# Peer review of "Accurate Detection of Salmonella Based on Microfluidic Chip to Avoid Aerosol Contamination"

_foods, 2022, doi:10.3390/foods11233887_

Round 1

Reviewer 1 Report

This study is a very good one. In addition, it is scientifically sound and has contributed admirably to scientific knowledge especially in the area of food-borne pathogen detection and control.   The microfluidic chip that was developed and integrated to LAMP and CRISPR/Cas 12a to detect Salmonella is an excellent innovation. More so, with a sensitivity and specificity of 100% and the ability of the method to detect Salmonella at 4h in Salmon-spiked with 30 cfu/25g, there is no doubt that the method is highly recommendable. This is, in addition to the fact that the system not only provides a highly sensitive end-point fluorescent visual detection of food-borne pathogens but also greatly eliminates the risk of aerosol contamination. This is great scientific finding. However I expected the great findings to be followed by very good presentation. The discussion which was very highly limited was barely visible in the results section. A work like this needs a good discussion so that the scientific inputs that led to the very innovative outcomes will be more apparent.   As I stated earlier, the study is high grade but will be much better if it is discussed more, especially   in line with the author's guide as specified by the Journal.

Author Response

Response to Reviewer 1

Point 1: The discussion which was very highly limited was barely visible in the results section. A work like this needs a good discussion so that the scientific inputs that led to the very innovative outcomes will be more apparent. As I stated earlier, the study is high grade but will be much better if it is discussed more, especially in line with the author's guide as specified by the Journal.

Response 1: Thanks for your professional suggestion! We have added “Discussion” in the 3.6 (Pages: 11-12, lines: 307-333).

  Reviewer 2 Report

The manuscript provides information about a microfluidic chip integrated with LAMP and CRISPR/Cas12a system for Salmonella detection. It is an interesting manuscript and I found some parts of it that the authors should clarify.

1. Introduction

ü    Page 2, lines 63-70: eliminate: “LAMP reaction solution was injected into the chamber through one channel. When the amplifi-  cation was completed, the CRISPR/Cas12a reaction solution was injected subsequently to mix with amplicons through another channel for fluorescent signal production. The risk of aerosol leakage was eliminated in this process owing to the leakproofness of the chip, hereby protecting the detection environment and reagents from aerosol pollution. With this chip, the false-positive results are greatly avoided, providing a platform for more accurate detection results in subsequent detection.” It is detailed in other parts of the manuscript.

3. Results 

ü    Page 7, lines 220-225: “Typically, the limit of detection is determined by setting a threshold, which can be calculated from the average fluorescence intensity and standard deviation of the negative control. The value of threshold equals the average intensity and three times the standard deviation [28]. When the detected value is greater than the threshold, the corresponding Salmonella concentration can be regarded as the detection limit”. Change to materials and methods section because it is not a result.

ü    Page 9, lines 246-248: eliminate “One Salmonella standard strain, 12 Salmonella clinical strains, nine non-Salmonella 246 standard strains, and three non-Salmonella clinical strains were selected for experiments 247 in this study to determine the sensitivity and specificity of this method”. Repeat text from materials and methods.

Discussion

            There is not any discussion in the manuscript.  

Table

ü    Page 2, Table 1. Change “subspecies name” to “bacteria”. Eliminate “abbreviation” column.

Author Response

Response to Reviewer 2

Point 1: Page 2, lines 63-70: eliminate: “LAMP reaction solution was injected into the chamber through one channel. When the amplification was completed, the CRISPR/Cas12a reaction solution was injected subsequently to mix with amplicons through another channel for fluorescent signal production. The risk of aerosol leakage was eliminated in this process owing to the leakproofness of the chip, hereby protecting the detection environment and reagents from aerosol pollution. With this chip, the false-positive results are greatly avoided, providing a platform for more accurate detection results in subsequent detection.” It is detailed in other parts of the manuscript.

Response 1: Thanks for your important suggestion! Yes, this part is repetitive, so we have eliminated it.

Point 2: Page 7, lines 220-225: “Typically, the limit of detection is determined by setting a threshold, which can be calculated from the average fluorescence intensity and standard deviation of the negative control. The value of threshold equals the average intensity and three times the standard deviation [28]. When the detected value is greater than the threshold, the corresponding Salmonella concentration can be regarded as the detection limit”. Change to materials and methods section because it is not a result.

Response 2: Thanks for your reminding. We have transferred this part to Material and methods section 2.6 (Page: 4, lines: 134-137).

Point 3: Page 9, lines 246-248: eliminate “One Salmonella standard strain, 12 Salmonella clinical strains, nine non-Salmonella 246 standard strains, and three non-Salmonella clinical strains were selected for experiments 247 in this study to determine the sensitivity and specificity of this method”. Repeat text from materials and methods.

Response 3: Thanks. We have deleted it.

Point 4: There is not any discussion in the manuscript.

Response 4: Thanks for your reminding. We have added “Discussion” in 3.6 (Pages: 11-12, lines: 307-333).

Point 5: Page 2, Table 1. Change “subspecies name” to “bacteria”. Eliminate “abbreviation” column.

Response 5: Thanks. We have changed “subspecies name” to “bacteria” and eliminated “abbreviation” column in Table 1 (Pages: 2-3, lines: 84-85).

Reviewer 3 Report

Comments on:

Accurate detection of Salmonella based on microfluidic chip to avoid aerosol contamination

The authors developed a microfluidic chip to integrate Loop-mediated isothermal amplification (LAMP) and Clustered regularly interspaced short palindromic repeats (CRISPR)/Cas12a system to detect Salmonella accurately. The study topic is interesting. The experimental section is well-designed and conducted. The results were properly analyzed and interpreted, as well as the conclusion is supported with data. However, I still have some concerns related to the research design, and the English language still has some grammatical and syntax errors.

Salmonella is more commonly detected in poultry. So, the application of this method in poultry meat samples shows more reality.

Spike and recovery experiment in real samples is suggested in this new method to check the effect of the matrix of real samples on the test efficiency in comparing with buffer solutions.

In real samples, usually contamination happens with a mixture of bacterial strains. So, why you did not test the specificity of your approach in case of the presence of target bacteria and non-target bacteria?

Abstract:

“Furthermore, this method can be used to detect Salmonella after enrichment for 4 h in salmon spiked with 30 CFU/25 g”. Do you mean was used, or can be used? I guess you already did.

Methods:

Some parts of the methods need to be clearly described for reproducibility.

Discussion:

It needs to be improved. You should interpret your results.

Author Response

Response to Reviewer 3

Point 1: The English language still has some grammatical and syntax errors.

Response 1: Thanks for your concern! We have commissioned ShineWrite.com (English editing company) to revise the English language of the manuscript.

Point 2: Salmonella is more commonly detected in poultry. So, the application of this method in poultry meat samples shows more reality.

Response 2: Yes, Salmonella is more commonly detected in poultry, and the application of this method in poultry meat samples shows more reality. So, we have valuated this method in chicken sample (Page: 5, Lines: 153-163). The results (Figure 5B) showed this method performed well in chicken sample (Page: 9, Lines: 265-278).

Point 3: Spike and recovery experiment in real samples is suggested in this new method to check the effect of the matrix of real samples on the test efficiency in comparing with buffer solutions.

Response 3: Yes, we agree that a new method should be evaluated in real samples and buffer solutions. In section 2.8, this method was evaluated in TSB medium, salmon sample and chicken sample by spiking with Salmonella (4×101, 4×102, 4×103 and 4×104 CFU/mL) in all three matrixes (Page: 5, Lines: 164-169). The results (Figure 6) showed this method had a stable detection capability in real samples in comparing with buffer solution (Page: 10, Lines: 283-289).

Point 4: In real samples, usually contamination happens with a mixture of bacterial strains. So, why you did not test the specificity of your approach in case of the presence of target bacteria and non-target bacteria?

Response 4: Yes, this method should be evaluated in real samples which contained target bacteria and non-target bacteria. In section 2.8 (Page: 5, lines: 153-163), we have evaluated the specificity of this method in real samples with salmon and chicken samples that were not artificially contaminated with Salmonella for detection. And the results (Figure 5 and Figure S3) showed this method had high specificity in real samples (Page: 9, lines: 265-278).

Point 5: “Furthermore, this method can be used to detect Salmonella after enrichment for 4 h in salmon spiked with 30 CFU/25 g”. Do you mean was used, or can be used? I guess you already did.

Response 5: Thanks for your expertise reminding! We have changed “can be used” to “was used” (Page: 1, line: 23).

Point 6: Some parts of the methods need to be clearly described for reproducibility.

Response 6: Thanks for your suggestion. We have added more detailed description of the method in 2.6 (Page: 4, lines: 134-139) and 2.8 (Page: 5, lines: 153-169).

Point 7: It needs to be improved. You should interpret your results.

Response 7: Thanks for your professional suggestion. We have added “section 3.6 Discussion” to interpret the results (Pages: 11-12, lines: 307-333).